# Indicators of Targeted Physical Fitness in Judo and Jujutsu—Preliminary Results of Research

**DOI:** 10.3390/ijerph18084347

**Published:** 2021-04-20

**Authors:** Wojciech J. Cynarski, Jan Słopecki, Bartosz Dziadek, Peter Böschen, Paweł Piepiora

**Affiliations:** 1Institute of Physical Culture Studies, University of Rzeszów, 35-959 Rzeszów, Poland; ela_cyn@wp.pl (W.J.C.); bdziadek@ur.edu.pl (B.D.); 2Idokan Poland Association, 35-959 Rzeszów, Poland; slopecki_jan@onet.eu; 3World Alliance of Martial Arts, 28197 Bremen, Germany; peter.boschen@gmail.com; 4Faculty of Physical Education and Sport, University School of Physical Education in Wrocław, 51-612 Wrocław, Poland

**Keywords:** martial arts, combat sports, physical fitness, dexterity, handgrip

## Abstract

(1) Study aim: This is a comparative study for judo and jujutsu practitioners. It has an intrinsic value. The aim of this study was to showcase a comparison of practitioners of judo and a similar martial art jujutsu with regard to manual abilities. The study applied the measurement of simple reaction time in response to a visual stimulus and handgrip measurement. (2) Materials and Methods: The group comprising *N* = 69 black belts from Poland and Germany (including 30 from judo and 39 from jujutsu) applied two trials: “grasping of Ditrich rod” and dynamometric handgrip measurement. The analysis of the results involved the calculations of arithmetic means, standard deviations, and Pearson correlations. Analysis of the differences (Mann–Whitney *U* test) and Student’s *t*-test were also applied to establish statistical differences. (3) Results: In the test involving handgrip measurement, the subjects from Poland (both those practicing judo and jujutsu) gained better results compared to their German counterparts. In the test involving grasping of Ditrich rod, a positive correlation was demonstrated in the group of German judokas between the age and reaction time of the subjects (r_xy_ = 0.66, *p* < 0.05), as well as in the group of jujutsu subjects between body weight and the reaction time (r_xy_ = 0.49, *p* < 0.05). A significant and strong correlation between handgrip and weight was also established for the group of German judokas (r_xy_ = 0.75, *p* < 0.05). In Polish competitors, the correlations were only established between the age and handgrip measurements (r_xy_ = 0.49, *p* < 0.05). (4) Conclusions: Simple reaction times in response to visual stimulation were shorter in the subjects practicing the martial art jujutsu. However, the statement regarding the advantage of the judokas in terms of handgrip force was not confirmed by the results.

## 1. Introduction

Combat sports involving wrestling have their special characteristics, which include both performance training, specialist training, and teaching of technical–tactical skills. This applies to both training of judo as well as jujutsu as a sport discipline [1]. Both these disciplines have their sources in the old Japanese martial art jujutsu. However, due to the different objective sought in these disciplines, the results of the training involving the practitioners of sport jujutsu and traditional jujutsu assumes different directions [2]. The practitioners of the traditional martial arts jujutsu should be capable of combat in actual conditions of self-defense in the situation of a direct attack of several opponents on one and in various non-standard circumstances. In turn, judo forms a sport in which an emphasis is placed on the athletic preparation. In both cases (both in combat sports and martial arts), it is necessary to develop a high level of motor coordination skills [3,4,5]. The comprehensive array of these issues is best explained by the General Theory of Fighting Arts [6,7]. The *modern* jujutsu analyzed here is practiced as a traditional martial art and does not involve sport competition. However, both judo and jujutsu are open skill sports [8].

The authors of this paper adopted certain initial assumptions imposed by the needs of the study into motor skills in humans. The motor skills are therefore understood broadly. The authors assumed that the specific motor skills and the ability to perform specific motor activities are determined by the structure and functions of the organism. These motor abilities can be identified and developed as part of the structure and functions of the living organisms [6,9,10,11,12,13,14,15,16]. Among the various motor skills, we can distinguish skills that are especially important for particular sports.

The “targeted physical fitness” is the targeted physical fitness for judo and jujutsu. It involves the development of “manual skills” or “manual abilities” that determine the effectiveness of technical-tactical activities. Most of the fighting techniques in both fighting arts are so-called hand techniques (grips, throws, etc.).

The aim of the research was to compare two groups of subjects training in fighting arts, judo, and related jujutsu, with regard to manual capabilities. On the basis of long-term participant observation in the environment of the practitioners of sport judo and martial arts jujutsu, the authors collected data by performing observations and taking notes. The applicability of selected tests for research results from the experience gained by these authors. The focus in the study was placed on the phenomena (abilities) characteristic for the subjects practicing the martial arts jujutsu and the sport judo. On the basis of their experiences, the authors decided to apply a passive participant observation so as to be able to select the adequate abilities applied in the testing and analysis. The perception of these dependencies is not incidental but was performed in a continuous and systematic manner. Judo and jujutsu are both grappling sports. Most (in judo, almost all) techniques are performed in the grip. Therefore, the grip strength is a very important parameter here. On the basis of the above assumptions, we developed the main hypothesis and two detailed hypotheses. The main hypothesis was assumed in the form: there are differences in the motor abilities of practitioners of the martial art of jujutsu and judo. The detailed hypotheses assume that

Simple reaction time in response to a visual stimulus is shorter in the subjects practicing martial arts jujutsu.Handgrip in practitioners of judo is greater than in the ones practicing martial art jujutsu.

The research results are useful for trainers and researchers, presenting a tool that is easy to apply in the conditions of a training room and a short training unit time. The ease of testing strength and dexterity and their connection with “grappling” sports constitute a valuable resource for the practice of sports training.

## 2. Material and Methods

The study purposefully selected subjects with a similar training experience in judo and jujutsu in the range from 10 to 30 years, with a note that all those involved in the study have master ranks in their disciplines (a minimum of 1 dan); the age range of the subjects was 18 to 45 years. For the purpose of reliability of the results, the group selected for the present study had to be carefully selected. In the case of jujutsu, they were people more active in the study of this martial art—participants of international training seminars. In judo, only activity and sports results enable the achievement of higher technical degrees. Therefore, the selection concerned people with higher grades (black belts). The purposeful selection carried out in this way can be considered representative here. As a result, the group comprising Polish and German practitioners was selected, *N* = 69 in total, including 40 subjects from Poland (20 judoka and 20 jujutsuka) and 29 from Germany (10 judoka and 19 jujutsuka).

The group of Polish jujutsu was represented by 20 *jujutsuka*, all members of the “Budokan Poland” team. The subjects practicing martial arts jujutsu often practice it for several dozen years, which has a big impact on the level of physical coordination in these people, and therefore we decided to perform a comparison using a similar group of study subjects (on the basis of training experience, age, and dan level) practicing judo in two clubs located in Warsaw—University of Warsaw and Warsaw University of Technology, comprising 20 judokas in total.

The group originating from Germany was made up of participants of the following training seminars: “Allen Sally Seminar” in Delmenhorst (19 November 2017), “ACS meets Friends Seminar” in Bremen (10 March 2018), and “Ronnin Ju-Jitsu” in Wolfenbüttel (11 March 2018). The black belts in judo comprised the subjects who attended seminars in Bremen and Wolfenbüttel—10 people in total. The remaining 19 people were representatives of jujutsu and related martial arts. They train 3 times a week in each group.

### 2.1. Methods and Technique Applied in Study

Coordination ability and reaction time seem to be dominant and characteristic of jujutsu practitioners. In turn, among judokas, the handgrip may be of particular importance, which is used, for example, in the *keikogi* grip.

Selected motor and psychomotor abilities, which are characteristic and dominant among subjects practicing the martial art such as judo and jujutsu, were chosen to measure and compare. The study applied testing of the reaction force performed by grasping of Ditrich rod and a dynamometric handgrip measurement. The selected scope of the testing is characteristic for each of these disciplines, and their selection is closely related to the type of data that are recorded and focused on in the anticipated results. The knowledge about the characteristics of the subjects (such as age, weight, training experience, attained master rank) and the respective measurements that are carried out have a direct effect on the course of the statistical analysis. The study applied the measurements of grip strength and reaction time, capabilities and behaviors of the individuals, combined with in-depth knowledge of the various complex motor skills resulting from variable ontogenetic and environmental circumstances, and related to the specific characteristics of the practiced discipline.

The grasping of Ditrich rod is often applied and recommended for analytical testing in combat sports. It involves a test in which the reaction speed is measured, that is, an extremely important aspect of movement coordination in the art of jujutsu combat. The description of the testing procedure is as follows: The subject sits astride a chair, facing the rest, on which he places his forearm (resting it halfway down); four fingers are straightened and tightened, and the thumb is abducted. The tester holds a stick with a diameter and length of 50 cm, on which a centimeter scale is marked along its entire length. The lower end of the cane (0 cm) is at the level of the lower edge of the patient’s hand, approximately 1 cm from his hand. The tester lets go of the cane at any time. The subject’s task is to grasp the cane by clenching the hand. The distance from point 0 to the grip point (bottom edge) is measured. The present experiment followed the procedure developed by Ditrich [17]. The study subjects performed the procedure 5 times, and 2 extreme results were rejected. The arithmetic mean was calculated from the remaining trials.

The test involving the dynamometric measurement of hand force was carried out as follows: Dynamometer model: KERN MAP 130K1 palm hand dynamometer was used. The subject squeezes the hand dynamometer with their stronger hand. The wrist should lie in the extension line of the forearm. During the test, the test hand must not touch any part of the body. The strength of the hand is measured in kilograms. The better measurement of the 2 tests was selected for further analysis. The dynamometer should be adjusted to the size of the hand of the subjects so that the more distal finger joints fit in its handle. Hand swings during measurement are not allowed, because they can alter the results. Subjects need to focus mentally on the task, since the goal was to perform the measurement of the maximum handgrip force of the subjects.

### 2.2. Statistical Methods

In order to carry out analyses, taking into account the country of origin and the fighters’ style, for hand grip strength and Ditrich rod reaction time, we examined the normality of distribution in groups (Shapiro–Wilk test). Therefore, to test the significance of differences between the groups for the analyzed variables (handgrip, Ditrich’s measures), we used appropriate tests (*t*-test, UMW test—Mann–Whitney *U* test), and their effect size was given. The dominant hand was always examined.

All analyses were performed using the Statistica 13.3 software [18]. The R programming language with additional packages [19] was used to present the results in the form of graphs.

## 3. Results

The characteristics of the research group are presented in Table 1. The results of the tests performed showed that only reaction time (Ditrich’s measures) had a distribution close to normal.

The mean values obtained by the subjects from each group in the tests are presented in Figure 1 and Figure 2. Analyzing the average results characterizing hand strength (Figure 1), we found that the Polish competitors practicing judo (91.5 ± 9.05 kg) had the greatest hand strength, and the Germans practicing jujutsu had the weakest (average hand strength was 74.11 ± 12.00 kg). Hand grip strength among judokas and jujutsu practitioners significantly differed by country of origin (Figure 1).

On the basis of the average results obtained in the reaction speed test (Figure 2), we observed the best results among Polish jujutsu participants (average 97.2 mm), and the weakest among judo participants (Poland). Additionally, the average results of Polish players training in different fighting styles differed statistically significantly at the level of α = 0.05 (Figure 2).

For the variable characterizing the reaction speed of the subjects, we also performed a two-factor analysis of variance in the 2 × 2 scheme. Its aim was to investigate the influence of factors related to the origin of fighters (country) and the martial arts training style on the variability of reaction speed. The analysis of variance (Table 2) showed statistical significance for the main effect of country (*p* = 0.0499, η^2^ = 0.06) and the interaction of independent variables (country * martial arts—*p* = 0.0400, η^2^ = 0.06). The main effect related to the origin of the players indicated statistically significant differences in the average reaction speeds between Polish and German players.

Subsequently, the individual simple effects were analyzed, as shown in Figure 3. On the basis of the interaction graph, we found that there was a simple country effect in the judo group, where the respondents from Poland obtained weaker results than the respondents in the German group. Additionally, a simple effect of the origin of the respondents in the group of jujutsu practitioners was observed. Here, in turn, the Polish players were characterized by a better reaction speed than the players from Germany. The graph also showed a simple effect of the fighting style in the Polish group, where the jujutsu practitioners obtained better average reaction speed results than the judo practitioners. When analyzing the simple effect of martial arts in the German competitors’ group, we found comparable average levels of reaction speed of competitors practicing judo and jujutsu.

For the collected statistical material, the influence of the variables age, body weight, and competitor training experience on the results obtained in the tests of hand strength and reaction speed was also examined. On the basis of the results obtained (Table 3), we found that there were statistically significant correlations between the analyzed variables. Moreover, the positive direction of the relationship between the analyzed variables and the results of the Ditrich stick test indicated the higher the value of the tested feature, the slower the player’s reaction speed. In the case of measuring hand strength, the positive value of the coefficient indicated a directly proportional influence of the variable on the test result. The strongest relationships were established among German judo players. In this group, the hand strength obtained in the test was significantly influenced by the weight of the respondents (r_xy_ = 0.75), while the variable age had a significant influence on the reaction speed (r_xy_ = 0.66). In another group of players (Germany, jujutsu), the relationship between the weight of the subject and the speed of reaction also turned out to be significant. A statistically significant relationship between age and hand strength among Polish judo players was observed (r_xy_ = 0.49—Poland, judo).

## 4. Discussion

The issue related to the handgrip force and manual abilities concerns almost the entire range of the related combat sports, including judo, sambo, and wrestling. The central consideration seems to lie in the handgrip, which forms an important parameter to determine the abilities in wrestling sports [16,20]. In contrast, the techniques and tactics adopted in the combat of the traditional jujutsu are similar to *aikido*, and dissimilar from judo, with a note that the range of the available techniques also includes the category named *atemi kogeki waza*, which includes strikes and kicks (performed in various positions and distances) [1]. As was stated in similar research, “Contribution of grip strength in system permits to consider them important for successfulness in wrestling, judo etc. At the same time in kicking martial arts these indicators are not very important and their absolute contribution in system formation is much lower” [16]. The strong grip is particularly important in such combat sports, such as judo, sambo, wrestling, and MMA [21,22].

In the case of judo and sports jujutsu, special fitness tests are often used, on the basis of acceptable throws in sports combat [23]. Intermediate performance tests (reaction speed and manual efficiency), such as the Batak Lite test [15,24] or the balance test, as an element of the motor coordination ability, have an indirect effect on the combat effectiveness [25,26,27,28]. The test involving gripping of Ditrich rod deserved its role in the sports that require fast reaction, such as in fencing [29,30]. The level of manual abilities is often examined as well. This tool can be applied in sports and movement rehabilitation—namely, in the tests involving speed of reaction and hand–eye coordination [12,31,32]. However, we are familiar with the dependencies that occur between the type of the training and the level of various motor abilities. To provide an insight from literature, “the training period with intensive judo-specific drills that engage cognitive functions and require hand’s maximal static efforts improve psycho-motor ability but may impair hand grip strength” [6]. On the example of judo, it was also stated that “the use of standardized, specific tests seems to be a diagnostic tool for the rating of psycho-motor abilities in judo contestants” [20], where the handgrip force is one of indicators.

The level of motor abilities forms an important component of the positive health-related potential and is known to affect the level of the subjectively understood well-being. Hence, its role in sports cannot be underestimated. In particular jujutsu is commonly taken up for the reasons associated with the ability of self-defense as well as for recreation [4]. In addition, it can provide individuals with a feeling of personal security.

The way to gain the master rank in martial arts is not related to the results of sport competition; however, it is known to depend primarily on the involvement of the individuals and on various environmental and institutional determinants [5,7]. Therefore, it seems less important to analyze whether the study subjects come from Poland or Germany, and instead to assess their level of overall activity. Thus, the participants of seminars in martial arts (subjects from Germany) could form a group with a more active lifestyle. In addition, judokas from Germany participated in seminars concerned mainly with the practice of jujutsu. Therefore, their very good results in the “Ditrich rod” test (manual efficiency and speed of reaction) could be slightly better than judokas from Poland. Would it be a recommendable complementary sport for judokas? Perhaps it would be useful for coaches.

## 5. Conclusions

The results of the analysis allowed the authors to formulate the following conclusions:Simple reaction time in response to a visual stimulus is shorter in jujutsu practitioners (*p* = 0.003).There were no statistically significant differences between jujutsu and judo players in the dynamometric handgrip measurement.The two-factor analysis of variance showed a statistically significant effect of the interaction of country and martial arts variables on the results obtained by the respondents in the reaction speed test. Moreover, regardless of the type of martial arts, the average results obtained in the test with the Ditrich rod was significantly different between the country of origin of the participants.The Pearson’s linear correlation values determined for the studied groups indicate a negative impact of ageing players on their reaction speed. The weight of the players also had a negative influence on the reaction speed.The weight and age of the competitors (especially in judo) had a significant (*p* < 0.05) influence on the results obtained in the dynamometer measurement of hand strength.

With the help of these two tools, we were able to establish the difference in the result in favor of people practicing jujutsu (which is a more comprehensive martial art according to the technical repertoire). The simple research tools of the Ditrich rod and hand grip dynamometer can still be very useful in the practice of sport, particularly in the fighting arts analyzed here. Ease of use is an advantage here.

## Figures and Tables

**Figure 1 ijerph-18-04347-f001:**
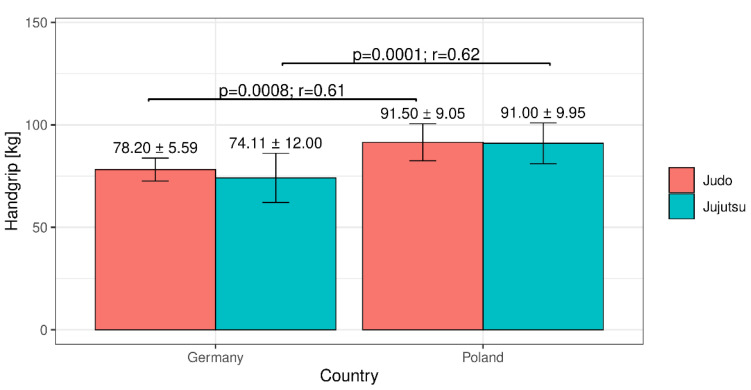
Mean values (±SD) obtained in the hand strength dynamometric test.

**Figure 2 ijerph-18-04347-f002:**
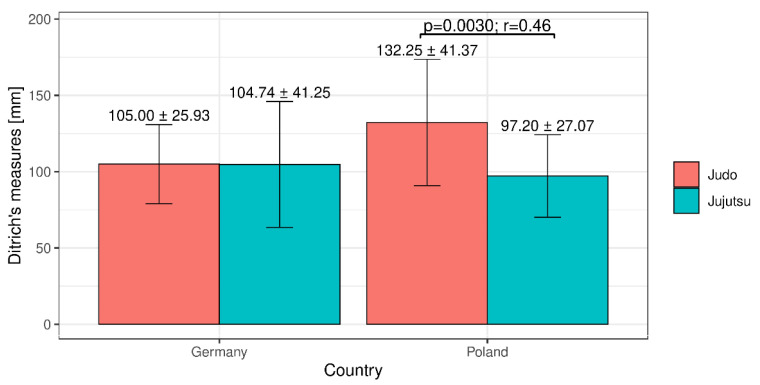
Mean values (±SD) obtained in the reaction speed test (Ditrich rod).

**Figure 3 ijerph-18-04347-f003:**
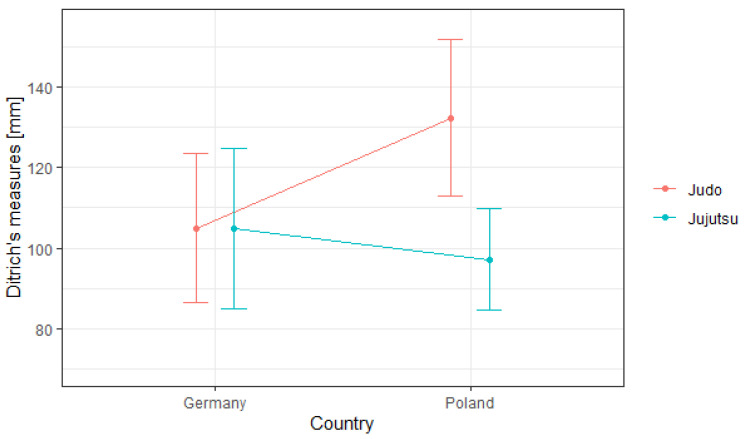
Simple effects interaction chart for country * martial arts effect.

**Table 1 ijerph-18-04347-t001:** Basic characteristics of sample.

Variables	*p*	Judo		Jujutsu		*p*
		Poland	Germany	Poland	Germany	
*N*		20	10	20	19	
Age (years) ^1^	0.3434	33.3 ± 8.03	30.1 ± 4.46	33.25 ± 7.29	45.42 ± 7.17	0.0001 *
Weight (kg) ^1^	0.7747	83.45 ± 14.22	86.4 ± 12.66	85.6 ± 11.78	95.68 ± 13.94	0.0151 *
Years of training (years) ^1^	0.7738	16.25 ± 6.87	15.7 ± 2.91	15.7 ± 3.96	28.84 ± 6.83	0.0001 *
Dominant hand, *n* (%)						
Left	----	----	----	----	----	----
Right		20 (100)	10 (100)	20 (100)	19 (100)	
Training experience (dan), *n* (%)						
1		18 (90.00)	8 (80.00)	19 (95.00)	7 (36.84)	
2	0.3540	1 (5.00)	2 (20.00)	1 (5.00)	4 (21.05)	0.0179 *
3		1 (5.00)	----	----	3 (15.79)	
4–10		----	----	----	5 (26.31)	

^1^—mean ± SD, *p*—probability obtained in the UMW test (χ^2^—for training experience variable); * statistical significance at the 0.05 level.

**Table 2 ijerph-18-04347-t002:** The result of the two-way analysis of variance for the variable reaction speed.

Effect	F	*p*	η^2^
Country	3.9896	0.0499 *	0.06
Martial arts	0.7075	0.4033	0.01
Country * martial arts	4.3881	0.0400 *	0.06

* Statistical significance at the 0.05 level.

**Table 3 ijerph-18-04347-t003:** The results of the Pearson linear correlation analysis for the studied variables.

	Variables	Poland		Germany	
		Handgrip	Ditrich’s Measures	Handgrip	Ditrich’s Measures
Overall	Age (years)	0.31 *	0.04	−0.14	0.26
	Weight (kg)	0.39 *	−0.10	0.13	0.35
	Years of training	0.17	0.12	−0.14	0.07
Judo	Age (years)	0.49 *	−0.19	0.48	0.66 *
	Weight (kg)	0.42	−0.26	0.75 *	−0.02
	Years of training	0.17	0.15	0.39	0.49
Jujutsu	Age (years)	0.14	0.43	−0.07	0.36
	Weight (kg)	0.36	0.28	0.09	0.49 *
	Years of training	0.17	0.02	−0.04	0.06

* Statistical significance at the 0.05 level.

## Data Availability

The authors confirm that the data supporting the findings of this study are available within the article.

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
