# Peer review of "Indicators of Targeted Physical Fitness in Judo and Jujutsu—Preliminary Results of Research"

_ijerph, 2021, doi:10.3390/ijerph18084347_

Round 1

Reviewer 1 Report

  1. There are no information about utilitarian purposes of the research.
  2. The same data are on table 2 and figures (1 & 2).
  3. There are no marking units on figure 1.
  4. There are summary instead conclusions.
  5. References should be completed.

Author Response

Answers for Reviewers

Many thanks to both Reviewers for their comments and suggestions. Thanks to this, our article will be rethought and improved.

For Reviewer no. 1

  1. There are no information about utilitarian purposes of the research.

Answer: The research results are useful for trainers and researchers, presenting a tool that is easy to apply in the conditions of a training room and a short training unit time. The ease of testing strength and dexterity and their connection with "grappling" sports constitute a valuable resource for the practice of sports training.

Added to text – lines 73-76.

  1. The same data are on table 2 and figures (1 & 2).

Answer: The duplicate data have been removed form manuscript (table 2). The charts have been supplemented with additional data. All references to table 2 have been removed from the text (line: 169, 179).

  1. There are no marking units on figure 1.

Answer: The figure 1 has been changed.

  1. There are summary instead conclusions.

Answer: The conclusions has been rebuilt.

The conclusion was added that “The Ditrich stick and dynamometer, these simple research tools, can still be useful in the practice of sport, in particular in the fighting arts analysed here.”

  1. References should be completed.

Answer: Five items were added and everything was put in order.

Reviewer 2 Report

General comments

            This study looked at using reaction time (assessed via Ditrich rod test method)  and grip strength (assessed via handgrip dynamometer) to determine differences between judokas and jujutsu martial art practitioners of similar experience and expertise. Although the article has potential, the writing is difficult to follow.  Authors’ writing requires extensive proof reading and a re-write from an native English language speaker with experience of the subject area (motor control and martial arts).  That said, content of the work is not really motor skills, it is grip strength and reaction time. The rationale of the study is unclear, why do this study in these populations,the samples are not well described and not really indicative of ‘martial arts; let alone specific to judo, etc.

Specific comments

Abstract- Data need to be presented here. State the hypothesis more specifically if you are going to refer to them, and report your results (correlations, t-tests, etc). Clearly you need to be brief, but you can state actual results. Why is karate study talked about here? For example, ‘positive correlation was demonstrated in the group of German judokas between the age and reaction time of the subjects (r= 0.66, p < 0.05), and in the group of jujutsu subjects between body weight and the reaction time (r =0.49, p<0.05).

Introduction

Page 1, line 34-35, this sentence is very unclear and difficult to interpret. In addition, I am not sure the first two sentence are necessary, I recommend removing lines 34-36 and starting the paper with ‘Combat sports involving wrestling have their special characteristics,…’

Line 39, However, the reference for the previous sentence does not exist, or is a mistake, as you only have 28 references cited and this one is listed as 30.

Line 40, typo- these

Line 42, instead of courses, as it implies classes, perhaps directions/routes or trajectories?

Page 2, Lines 48-49, reference 10 is on post-concussive syndrome and reaction time, I don’t think it is correctly referenced here (General Theory of Fighting Arts?)

Line 49-50, this sentence is unclear.  Perhaps analyzed, not analyzes? And practiced (or practised) does not have a ‘z’.

Lines 57-58, The aim of the research was to compare two groups of subjects training in martial arts, judo and related jujutsu martial arts, with regard to manual capabilities.

Line 63, on the basis of their experiences, the authors decided to utilize a passive particpant observation…

Material and methods

Line 78, selected instead of applied

Line 82, adequately selected?  Not the right word, carefully selected? 

Page 3, line 11-114, you already stated what you are doing, just combine the previous paragraph with this sentence so you are not repeating yourself.

Be consistent with verb tenses and pronouns.  (Lines 128-129, their and he)

Line 129-132, some of the explanation of the Ditrich test are unclear. Perhaps a figure or picture of the  test would be helpful.

Line 134, word missing, either lets ­go the rod at any moment or releases or drops the rod at any moment

Line 137-138, The present experiment followed the procedure developed by Ditrich (e.g., Raczek et al.) [26].

Line145-146, the better measurement of the two tests selected for further analysis.

Line 146-147, the dynamometer was adjusted…

Results

Line 152, analyses

Line 157-158, In order to carry 156 out analyses, taking into account the country of origin and the fighters' style, for hand grip strength and Ditrich stick reaction time,…

Line 159-160, The results of the tests performed showed that only reaction time (Ditrich's measures) has a distribution close to normal.

Page 4, line 170, report SD with means here and on page 5 line 172.

Page 5, I suggest re-phrasing lines 172-174.  Hand grip strength among judokas and jujutsu practitioners significantly differed by country of origin.

Line 179-182, this section can be stated more cleanly as the data are already in the table.  

Line 187, 0.06

Line 188-190, awkwardly phrased

Page 6, line 203-205, could be stated more cleanly.

Page 7, line 210, were instead of was

Line 210-213, more specifics needed. Nd you need to state somewhere that the correlations you are discussing were significant (at the 0.05 level?).

Line 216, relationships, not unions

Lines 218-221, writing needs to be more clear.  Line 218, ‘Association with’ might be better word choices than ‘influence on’.

Discussion

Line 234-235, particularly is misspelled.

Line 240-241, The article on fencing (and reference) seems out of place here.

Line 247-249, Poorly stated.  One example in judo also stated that…

Line 251-252, reference for the first sentence?

Page 8, line 259-260, rephrase:  … less important to analyze whether the study subjects come from Poland or Germany, and involvement plays a much greater role more – _that is their level of overall activity.

…less important to analyze whether the study subjects come from Poland or Germany, and instead assess their level of overall activity.

Conclusion

This entire section is very difficult to follow, so most of information is unclear.

Table 1, Title- Basic characteristics of sample

Age needs units of measure (yrs)

All participants are right-handed?  Unusual.

Your #2 notation under the table, I would move into the table.  So, Dominant hand, n(%) and Training experience, n(%).

Table 2, Title- Mean values (± SD) obtained in the individual tests.

Table 3- Effect spelled incorrectly

As I’m not sure if all versions will show the differently coloured ink, perhaps a note and an asterisk to indicate significant effects?

Table 4, Age needs units of measure(yrs)

Were these correlations significant at the 0.05 level?  State that below the table.

Figure 1- it is unclear to what the statistics on the figure refer.

Handgrip needs units of measure on the axis

Font size is different between title and figure.

Figure 3-A more complete title would be helpful: Simple effects interaction chart for Country * Martial Arts effect in Ditrich’s test of reaction time.

Reference list:

#18, word missing in title of article.  ‘Can physical tests predict the technical-tactical…’

#19, line 328, validity is misspelled

Author Response

Answers for Reviewers

Many thanks to both Reviewers for their comments and suggestions. Thanks to this, our article will be rethought and improved.

For Reviewer no. 2

General comments

            This study looked at using reaction time (assessed via Ditrich rod test method)  and grip strength (assessed via handgrip dynamometer) to determine differences between judokas and jujutsu martial art practitioners of similar experience and expertise. Although the article has potential, the writing is difficult to follow.  Authors’ writing requires extensive proof reading and a re-write from an native English language speaker with experience of the subject area (motor control and martial arts).  That said, content of the work is not really motor skills, it is grip strength and reaction time. The rationale of the study is unclear, why do this study in these populations, the samples are not well described and not really indicative of ‘martial arts; let alone specific to judo, etc.

Answer: The text was consulted with colleagues working in the field of motor control to improve the language.

The justification for the selection of the scientific problem and the selection of groups (from sports judo and traditional jujutsu) was justified in the Introduction.

Specific comments

Abstract- Data need to be presented here. State the hypothesis more specifically if you are going to refer to them, and report your results (correlations, t-tests, etc). Clearly you need to be brief, but you can state actual results. Why is karate study talked about here? For example, ‘positive correlation was demonstrated in the group of German judokas between the age and reaction time of the subjects (r= 0.66, p < 0.05), and in the group of jujutsu subjects between body weight and the reaction time (r =0.49, p<0.05).

Answer: Thank you very much for your suggestion. The abstract has been changed.

Introduction

Page 1, line 34-35, this sentence is very unclear and difficult to interpret. In addition, I am not sure the first two sentence are necessary, I recommend removing lines 34-36 and starting the paper with ‘Combat sports involving wrestling have their special characteristics,…’

Answer: Thank  you  very  much  for  your  suggestion. The lines 34-36 have been removed from manuscript.

Line 39, However, the reference for the previous sentence does not exist, or is a mistake, as you only have 28 references cited and this one is listed as 30.

Answer: The reference in line 39 has been corrected.

Line 40, typo- these

Answer: The word “theses” has been corrected to “these”.

Line 42, instead of courses, as it implies classes, perhaps directions/routes or trajectories?

Answer: The word “courses” has been changed to “directions”.

Page 2, Lines 48-49, reference 10 is on post-concussive syndrome and reaction time, I don’t think it is correctly referenced here (General Theory of Fighting Arts?)

Answer: The reference “10” was wrong. It has been corrected.

Line 49-50, this sentence is unclear.  Perhaps analyzed, not analyzes? And practiced (or practised) does not have a ‘z’.

Answer: Thank  you  very  much  for  your  suggestion. The words “analyses” and “practized” have been corrected to “analysis” and “practiced”.

Lines 57-58, The aim of the research was to compare two groups of subjects training in martial arts, judo and related jujutsu martial arts, with regard to manual capabilities.

Answer: Thank  you  very  much  for  your  suggestion. The sentence has been corrected.

Line 63, on the basis of their experiences, the authors decided to utilize a passive particpant observation…

Answer: Thank  you  very  much  for  your  suggestion. The sentence has been corrected.

Material and methods

Line 78, selected instead of applied

Answer: Corrected. The word “applied” has been changed to “selected”.

Line 82, adequately selected?  Not the right word, carefully selected?

Answer: Thank  you  very  much  for  your  suggestion. The term “adequately selected” has been replaced with the term “carefully selected”.

Page 3, line 11-114, you already stated what you are doing, just combine the previous paragraph with this sentence so you are not repeating yourself.

Answer: Thank you for your valuable comment. The text in line 105-114 has been changed.

The following sentence has been added: The coordination abilities and reaction time seem to be dominant and characteristic of jujutsu practitioners. In turn, among judokas, the handgrip may be of particular importance which is used, for example, in the keikogi grip.

In the study, it was decided to measure and compare selected motor and psychomotor abilities, that are characteristic and dominant among subjects practicing the martial art such as judo and jujutsu.

Be consistent with verb tenses and pronouns.  (Lines 128-129, their and he)

Answer: The word “their” has been changed to “his”.

Line 129-132, some of the explanation of the Ditrich test are unclear. Perhaps a figure or picture of the  test would be helpful.

Answer: Thank you for your comment. The explanation of the Ditrich test has been changed.

Line 134, word missing, either lets ­go the rod at any moment or releases or drops the rod at any moment

Answer: The word “releases” has been added.

Line 137-138, The present experiment followed the procedure developed by Ditrich (e.g., Raczek et al.) [26].

Answer: Thank  you  very  much  for  your  suggestion. The sentence has been changed.

Line 145-146, the better measurement of the two tests selected for further analysis.

Answer: Thank  you  very  much  for  your  suggestion. The sentence has been changed.

Line 146-147, the dynamometer was adjusted…

Answer: The term “should be” has been replaced with the term “was adjusted”

Results

Line 152, analyses

Answer: The word “analyzes” has been corrected to “analyses”.

Line 157-158, In order to carry 156 out analyses, taking into account the country of origin and the fighters' style, for hand grip strength and Ditrich stick reaction time,…

Answer: Thank  you  very  much  for  your  suggestion. The sentence has been changed.

Line 159-160, The results of the tests performed showed that only reaction time (Ditrich's measures) has a distribution close to normal.

Answer: Thank  you  very  much  for  your  suggestion. The sentence has been changed.

Page 4, line 170, report SD with means here and on page 5 line 172.

Answer: The values of SD have been added to means.

Page 5, I suggest re-phrasing lines 172-174.  Hand grip strength among judokas and jujutsu practitioners significantly differed by country of origin.

Answer: Thank  you  very  much  for  your  suggestion. The sentence has been changed.

Line 179-182, this section can be stated more cleanly as the data are already in the table. 

Answer: The duplicate data have been removed form manuscript (table 2). The charts have been supplemented with additional data from table 2. All references to table 2 have been removed from the text (line: 169, 179).

Line 187, 0.06

Answer: The value has been corrected.

Line 188-190, awkwardly phrased

Answer: The value has been corrected. The following sentence has been added:

The main effect related to the origin of the players indicates statistically significant differences in the average reaction speeds between Polish and German players.

Page 6, line 203-205, could be stated more cleanly.

Answer: Thank you for your suggestion. The sentence has been written more cleanly. 

Page 7, line 210, were instead of was

Answer: The word “was” has been replaced with the word “were”.

Line 210-213, more specifics needed. Nd you need to state somewhere that the correlations you are discussing were significant (at the 0.05 level?).

Answer: Information about the statistical significance assumed in the analysis (statistical significance at the 0.05 level) was added to the description of the tables.

Line 216, relationships, not unions

Answer: Thank you for your suggestion. The word “unions” has been changed to “relationships”.

Lines 218-221, writing needs to be more clear. 

Answer: Thank you very much for your suggestion. The text has been corrected.

Line 218, ‘Association with’ might be better word choices than ‘influence on’.

Answer: Corrected. The term “influence on” has been replaced with the term “association with”.

Discussion

Line 234-235, particularly is misspelled.

Answer: The word has been corrected.

Line 240-241, The article on fencing (and reference) seems out of place here.

Answer: Thank you for your comment. The article on fencing has been removed from the manuscript.

Line 247-249, Poorly stated.  One example in judo also stated that…

Answer: This is about some facts – in Discussion. However, it was changed “This tool is still commonly applied …” into: ”This tool can be applied …”.

Line 251-252, reference for the first sentence?

Answer: This reference concerns the sentence that precedes it.

Page 8, line 259-260, rephrase:  … less important to analyze whether the study subjects come from Poland or Germany, and involvement plays a much greater role more – _that is their level of overall activity.

…less important to analyze whether the study subjects come from Poland or Germany, and instead assess their level of overall activity.

Answer: Thank you very much for your suggestion. The text has been corrected.

Conclusion

This entire section is very difficult to follow, so most of information is unclear.

Answer: The conclusions have been rebuilt.

The results of the analyzes allowed to formulate the following conclusions:

  • Simple reaction time in response to a visual stimulus is shorter in jujutsu practitioners (p= 0.003).
  • There were no statistically significant differences between jujutsu and judo players in the dynamometric handgrip measurement.
  • The two-factor analysis of variance showed a statistically significant effect of the interaction of country and martial arts variables on the results obtained by the respondents in the reaction speed test. Also, the country of origin of the players statistically significantly differentiated, regardless of the type of martial arts, the average results obtained in the test with the Ditrich rod.
  • The Pearson's linear correlation values determined for the studied groups indicate a negative impact of ageing players on their reaction speed. The weight of the players also had a negative influence on the reaction speed.
  • The weight and age of the competitors (especially in judo) had a significant (p < 0.05) influence on the results obtained in the dynamometer measurement of hand strength.
  • The Ditrich rod and dynamometer, these simple research tools, can still be useful in the practice of sport, in particular in the fighting arts analysed here.

Table 1, Title- Basic characteristics of sample

Answer: The title has been changed.

Age needs units of measure (yrs)

Answer: Thank you for your suggestion. The unit of measure has been added.

All participants are right-handed?  Unusual.

Answer: In this group all participants were right-handed.

Your #2 notation under the table, I would move into the table.  So, Dominant hand, n(%) and Training experience, n(%).

Answer: Thank you for your suggestion. #2 notation has been moved into the table. 

Table 2, Title- Mean values (± SD) obtained in the individual tests.

Answer: Thank you for your suggestion. Table 2 has been removed from the manuscript. Based on your advice, we decided to change the titles of the figures.

Table 3- Effect spelled incorrectly

Answer: Corrected. The symbol of effect “Eta2p has been replaced with the symbol “h2”.

As I’m not sure if all versions will show the differently coloured ink, perhaps a note and an asterisk to indicate significant effects?

Answer: Thank you for your valuable comment. The asterisk has been added to the results indicating significant effects. To the description of the tables, the note has been added.

Table 4, Age needs units of measure (yrs)

Answer: The unit of measure has been added to Table 4 (now Table 3).

Were these correlations significant at the 0.05 level?  State that below the table.

Answer: Corrected. The note has been added below the table.

Figure 1- it is unclear to what the statistics on the figure refer.

Answer: Corrected. Figures 1–2 and the titles of these figures have been changed.

Handgrip needs units of measure on the axis

Answer: Corrected. The units have been added to the axis.

Font size is different between title and figure.

Answer: Thank you for your suggestion. The font was standardized.

Figure 3-A more complete title would be helpful: Simple effects interaction chart for Country * Martial Arts effect in Ditrich’s test of reaction time.

Answer: Thank you for your suggestion. The title of Figure 3 has been changed.

Reference list:

#18, word missing in title of article.  ‘Can physical tests predict the technical-tactical…’

Answer: Thank you for your suggestion. The title of the article has been corrected.

#19, line 328, validity is misspelled

Thank you for your suggestion. The misspelled word has been corrected.

The comments relate essentially to the precision of the utterance (logic and language). We have made the required corrections. Thank you for the suggestion of the indicated corrections. The reference to the publication on fencing (item 4 in the original manuscript) has been removed.

Reviewer 3 Report

The Article “Indicators of targeted physical fitness in judo and jujutsu - pre-2 liminary results of research” requires several changes before it will be published. The major strength of the study is this study practical application. However, there are some remarks concerning this article:

  1. The aim of this study is a comparison of practitioners of judo and a similar martial art jujutsu with 13 regard to manual abilities. It is not clear what does it mean “manual abilities”? Additionally, in the title of the article mentioned “targeted training”, it would be more valuable if it will be more explained in the introduction (is ir related to the training periodization, or to the purposeful training or etc.). As well as is important at least to mention what other physical components are important for the performance / competitive activities in judo or jujutsu.
  2. I would suggest calling in all the article physical fitness components names, but not to call them as a test names. Some terms used in the article not always correctly (e.g. “reaction rate test” – line 178).
  3. By the way, as a key word presented “dexterity” (even it is better to call in sport setting “agility”), but not any test or results related to that presented in the article.
  4. There are some parts in the article to much repetitive.
  5. It is not clear why it is necessary in this research include data form observations and taking notes? (e.g. lines 58-61).
  6. Additionally, in my opinion, it is strange to mention that selection of participants was considered as representative if it was just sportsmen form one sport club (line 90-91). Moreover, the number of the participants in the research in several places are presented not the same (line 16, line 87-88, line 98-100).
  7. In the material and methods part, I would suggest presenting all the material more systematically, because now it looks a little bit chaotically.
  8. All the tests presented without references. The explanation grasping Ditrich’s rod test presented without measured units. It is not clear was both tests done with dominant hand or not? I think both hands are very important in judo and jujutsu, therefore it would be more valuable to present data form both hands.
  9. Statistical analysis and used test for the comparison has to be explained in statistical methods, but not in the results part (e.g. lines 156-159).
  10. I would suggest into results include BMI as an indicator of body composition and try to find some relations to other fitness components.
  11. There is not in every place mentioned difference, although it could be (e.g. table 1 German jujutsu group weight in comparison to other results). Additionally in this table it is not clear presentation of training experience in dan, maybe it is even would be valuable to group data form 4 dan and below, because there only 5 subjects and just in German jujutsu group (if I understood correctly).
  12. Table 2 and figure 1 and 2 shoes data repetition.
  13. In Figure 3 title explains only one physical fitness component (reaction time) effect interaction for country and martial art. What about effect of dynamometry?
  14. In table 4 presented corelation without significance level, this is not also explained in the text. Link to the table presented below the table (e.g. line 210).
  15. There is some material presented in discussion part not really related to this discussed topic. 

Before publication, in my opinion, article must be improved.

Author Response

Answers for Reviewers

Many thanks to both Reviewers for their comments and suggestions. Thanks to this, our article will be rethought and improved.

For Reviewer no. 3.

The Article “Indicators of targeted physical fitness in judo and jujutsu - preliminary results of research” requires several changes before it will be published. The major strength of the study is this study practical application. However, there are some remarks concerning this article:

The aim of this study is a comparison of practitioners of judo and a similar martial art jujutsu with 13 regard to manual abilities. It is not clear what does it mean “manual abilities”? Additionally, in the title of the article mentioned “targeted training”, it would be more valuable if it will be more explained in the introduction (is it related to the training periodization, or to the purposeful training or etc.). As well as is important at least to mention what other physical components are important for the performance / competitive activities in judo or jujutsu.

Answer: An explanation has been added:

The "targeted physical fitness" is the targeted physical fitness for judo and jujutsu. It involves the development of “manual skills” or “manual abilities” that determine the effectiveness of technical-tactical activities. Most of the fighting techniques in both fighting arts are so-called hand techniques (grips, throws, etc.).

    I would suggest calling in all the article physical fitness components names, but not to call them as a test names. Some terms used in the article not always correctly (e.g. “reaction rate test” – line 178).

Answer: Thank you very much for your suggestion. The text has been corrected.

    By the way, as a key word presented “dexterity” (even it is better to call in sport setting “agility”), but not any test or results related to that presented in the article.

Answer: Indeed, the research carried out is about dexterity (which is indirectly checked by the Ditrich test) and the improvement of directed fitness.

    There are some parts in the article to much repetitive.

    It is not clear why it is necessary in this research include data form observations and taking notes? (e.g. lines 58-61).

Answer: Not without significance is the over 50-year participant observation of one of the co-authors in the judo and jujutsu community, who first as a competitor and then a trainer with a master's degree knows the specifics of both martial arts from the inside. This is important for a general interpretation of the issues raised here.

    Additionally, in my opinion, it is strange to mention that selection of participants was considered as representative if it was just sportsmen form one sport club (line 90-91).

Answer: The selection was deliberate, but made in such a way as to meet the criterion of representativeness, while maintaining the number of samples of the same order (however, they do not have to be equal).

Moreover, the number of the participants in the research in several places are presented not the same (line 16, line 87-88, line 98-100).

Answer: The number of participants has been checked. The description of groups has been written more clearly.

    In the material and methods part, I would suggest presenting all the material more systematically, because now it looks a little bit chaotically.

Answer: Thank you for your suggestion. Material and methods have been changed partially.

    All the tests presented without references. The explanation grasping Ditrich’s rod test presented without measured units.

Answer: The explanation for grasping Ditrich’s rod test has been corrected.

It is not clear was both tests done with dominant hand or not? I think both hands are very important in judo and jujutsu, therefore it would be more valuable to present data form both hands.

Answer: In this group all participants were right-handed. Both tests were done with the dominant hand. The other hand wasn't tested.

    Statistical analysis and used test for the comparison has to be explained in statistical methods, but not in the results part (e.g. lines 156-159).

Answer:  Thank you for your suggestion. Description of used statistical tests moved to statistical methods.

    I would suggest into results include BMI as an indicator of body composition and try to find some relations to other fitness components.

Answer: The BMI measurement, which gives a similar result of an athletic bodybuilder and an obese person, is omitted and only gives an idea of the leanness in body proportions.

    There is not in every place mentioned difference, although it could be (e.g. table 1 German jujutsu group weight in comparison to other results). Additionally in this table it is not clear presentation of training experience in dan, maybe it is even would be valuable to group data form 4 dan and below, because there only 5 subjects and just in German jujutsu group (if I understood correctly).

Answer: Thank you for your suggestion. Table 1 has been changed.

    Table 2 and figure 1 and 2 shoes data repetition.

Answer: The duplicate data have been removed form manuscript (table 2). The charts have been supplemented with additional data from table 2. All references to table 2 have been removed from the text (line: 169, 179).

    In Figure 3 title explains only one physical fitness component (reaction time) effect interaction for country and martial art. What about effect of dynamometry?

Answer: We didn't illustrate the interaction effect (Two-way ANOVA) for hand strength, because the dependent variable didn't meet the assumption of normal distribution in the groups under consideration.

    In table 4 presented correlation without significance level, this is not also explained in the text. Link to the table presented below the table (e.g. line 210).

Answer: Information about the statistical significance assumed in the analysis (statistical significance at the 0.05 level) was added to the description of the tables.

    There is some material presented in discussion part not really related to this discussed topic.

Answer: Among the co-authors, we selected the list of references, rejecting thematically related to a small extent (like items 4, 25, 38 of the original version), and adding better thematically correlated.

Before publication, in my opinion, article must be improved.

Answers: Thank you very much for your valuable comments and suggestions that helped to improve the content of this work.

Round 2

Reviewer 2 Report

The authors are to be commended for this much-improved version of the paper. There are still a few minor issues, see specific comments, but the language and structure are both much better.

Page 2 lines 48-52, this section is still not entirely clear.

Lines 54-57, different paragraph spacing and not consistent with the rest of the paper. Topic needs a transition, as it does not flow from previous section.

Lines 78-81, much better phrasing

Page 3, line 110,  Coordination ability and reaction time…

Line 113-114, Selected motor and psychomotor abilities, that are characteristic and dominant among subjects practicing the martial art such as judo and jujutsu, were chosen to measure and compare.

Lines 129-136, description of Ditrich method is much improved.

Page 4, line 143-144, word missing. The better measurement of the two tests was selected for further analysis

Page 5, line 178-179, it might be better stated as ‘jujitsu participants’ or judo participants’

Table 2, Efect is still spelled incorrectly: Effect

Shift note left under table

Table 3, shift note left under table

Page 6, line 209, it was found that… (apologies, I am not sure what I was looking at before, you were correct the first time).

Page 8, line 268 analysis:  The results of the analysis allowed the authors to formulate the following conclusions:

Line 275-277,  awkward phrasing. Suggestion:   Also, regardless of the type of martial arts, the average results obtained in the test with the Ditrich rod was significantly different between the country of origin of the participants.

Line 284-285, rephrase: The simple research tools of the  Ditrich rod and hand grip dynamometer can still be very useful in the practice of sport, in particular in the fighting arts analysed here.

The conclusion needs some sort of summary statement to tie it all together.  Perhaps expanding on the above sentence?

Newly added references need to be in consistent formatting with journal requirements.

Author Response

Authors' Response, round 2, paper “Indicators of targeted physical fitness in judo and jujutsu - preliminary results of research”

We would like to thank Editor and Reviewers for their comments that helped us to improve the manuscript. We greatly appreciate their comments and the time committed to evaluate our work. The entire manuscript has been rebuilt and changed as suggested.

Answers for Reviewer 2:

Page 2 lines 48-52, this section is still not entirely clear.

Answer: In the next paragraph, we move from a general discussion of motor skills to the issue of directed fitness. The development of individual issues can be found in the quoted bibliography. A linking sentence has been added.

Lines 54-57, different paragraph spacing and not consistent with the rest of the paper. Topic needs a transition, as it does not flow from previous section.

Answer: The spacing between paragraphs has been corrected.

In this paragraph, we explain what directed fitness is involved in judo, jujutsu or related martial arts.

Lines 78-81, much better phrasing

Answer: Thank you for your valuable comment.

Page 3, line 110,  Coordination ability and reaction time…

Answer: Thank you for your suggestion. The text has been corrected.

Line 113-114, Selected motor and psychomotor abilities, that are characteristic and dominant among subjects practicing the martial art such as judo and jujutsu, were chosen to measure and compare.

Answer: Thank you very much for your suggestion. The text has been changed.

Lines 129-136, description of Ditrich method is much improved.

Answer: Thank you very much for your valuable comment.

Page 4, line 143-144, word missing. The better measurement of the two tests was selected for further analysis

Answer: Thank you for your suggestion. The missing word has been added.

Page 5, line 178-179, it might be better stated as ‘jujitsu participants’ or judo participants’

Answer: The text in line 178-179 has been corrected. The word “players” replaced by the word “participants”.

Table 2, Efect is still spelled incorrectly: Effect

Answer: Thank you for your suggestion. The word has been corrected.

Shift note left under table

Answer: The note under the table 2 has been shifted to the left side.

Table 3, shift note left under table

Answer: The note under the table 3 has been shifted to the left side.

Page 6, line 209, it was found that… (apologies, I am not sure what I was looking at before, you were correct the first time).

Answer: The text has been corrected.

Page 8, line 268 analysis:  The results of the analysis allowed the authors to formulate the following conclusions:

Answer: Thank you for your suggestion. The sentence has been corrected.

Line 275-277,  awkward phrasing. Suggestion:   Also, regardless of the type of martial arts, the average results obtained in the test with the Ditrich rod was significantly different between the country of origin of the participants.

Answer: Thank you for your suggestion. The sentence has been corrected.

Line 284-285, rephrase: The simple research tools of the  Ditrich rod and hand grip dynamometer can still be very useful in the practice of sport, in particular in the fighting arts analysed here.

Answer: Thank you for your suggestion. The sentence has been rephrased.

The conclusion needs some sort of summary statement to tie it all together.  Perhaps expanding on the above sentence?

Answer: This has been corrected and supplemented - the suggested modification has been made.

Newly added references need to be in consistent formatting with journal requirements.

Answer: The bibliographic record has been corrected.
